# 3D-integrated multilayered physical reservoir array for learning and forecasting time-series information

Sanghyeon Choi[1,2,6], Jaeho Shin[1,3], Gwanyeong Park[1], Jung Sun Eo[1], Jingon Jang[1,7], J. Joshua Yang [2] ✉ & Gunuk Wang [1,4,5] ✉

A wide reservoir computing system is an advanced architecture composed of multiple reservoir layers in parallel, which enables more complex and diverse internal dynamics for multiple time-series information processing. However, its hardware implementation has not yet been realized due to the lack of a high-performance physical reservoir and the complexity of fabricating multiple stacks. Here, we achieve a proof-of-principle demonstration of such hardware made of a multilayered three-dimensional stacked $3 \times 10 \times 10$ tungsten oxide memristive crossbar array, with which we further realize a wide physical reservoir computing for efficient learning and forecasting of multiple time-series data. Because a three-layer structure allows the seamless and effective extraction of intricate three-dimensional local features produced by various temporal inputs, it can readily outperform two-dimensional based approaches extensively studied previously. Our demonstration paves the way for wide physical reservoir computing systems capable of efficiently processing multiple dynamic time-series information.

Big data has emerged as a critical driving force to the data-driven industrial revolution, which underpins decision-making processes in modern society, enhances efficiency, and promotes innovation[1,2]. The ability to collect, analyze, and interpret big data supports companies and societies involved in artificial intelligence in maintaining a competing edge. Among the various types of data, time-series data, which are sequential observations collected over specific time frames, have become increasingly important as dynamic information in applications such as biometric analysis[3], weather forecasting[4], stock chart estimation[5], and water inflow prediction[6]. Their importance arises from the inherently ordered nature, which allows predicting future behaviors in dynamic information using estimated models that describe correlated features.

In artificial intelligence, time-series data have been extensively explored through approaches including network, statistical, and transformer models[7,8]. A promising approach is the implementation of reservoir computing (RC) based on nonlinear dynamic systems such as physical reservoirs (PRs) to replace computationally intensive software-based methods[9,10]. Remarkably, memristive systems feature inherent nonlinearity and short-term memory capability, which enable a single memristor node to replace a complex nonlinearly coupled loop formed by multiple nodes in a software reservoir[11–18]. Since the

[1]KU-KIST Graduate School of Converging Science and Technology, Korea University, 145 Anam-ro, Seongbuk-gu, Seoul 02841, Republic of Korea. [2]Department of Electrical and Computer Engineering, University of Southern California, Los Angeles, CA 90089, USA. [3]Department of Chemistry, Rice University, 6100 Main Street, Houston, TX 77005, USA. [4]Department of Integrative Energy Engineering, Korea University, 145 Anam-ro, Seongbuk-gu, Seoul 02841, Republic of Korea. [5]Center for Neuromorphic Engineering, Korea Institute of Science and Technology, Seoul 02792, Republic of Korea. [6]Present address: Department of Electrical and Computer Engineering, University of California, Santa Barbara, CA 93106, USA. [7]Present address: School of Computer and Information Engineering, Kwangwoon University, 20 Kwangwoon-ro, Nowon-gu, Seoul 01897, Republic of Korea. ✉e-mail: jjoshuay@usc.edu; gunukwang@korea.ac.kr

first demonstration of memristive physical RC[11], several approaches based on memristors, such as the virtual node concept[14], self-organized nanowire network[17], input masking[16], and a fully analog framework[18], have been suggested to date. Although these proposed physical RC systems are noteworthy, most of them are fabricated on single-stacked and two-dimensional (2D) memristive device frameworks and inevitably restricted to a single reservoir architecture. The 2D-based approaches eventually encounter physical limitations in addressing multi-variant time information. In addition, several multi-layered physical RC systems have been recently proposed[19,20]. These systems demonstrated the promising aspects of the advanced reservoir architecture, including improved reservoir capacity and optimization. However, these approaches have a 2D lateral architecture or rely on the input design rule, which requires additional multilayered PR architecture capable of parallel processing, simplification, and versatility to efficiently address multi-variable temporal inputs (detailed in Supplementary Note 1). Hence, it is desirable to realize a wide physical RC with multiple-stacked PRs based on three-dimensional (3D) integrated memristive architecture. This 3D approach is anticipated to enable the extraction of richer features and multiple local traits from temporal inputs simultaneously, performing a more accurate, reliable, and rapid RC than the 2D-based single physical RC. In addition, the multiple-stacked PRs can provide signal mapping capabilities with a smaller footprint than a single PR. Consequently, 3D-integrated memristive PRs architecture is a significant step in the development of a next-generation physical RC. However, the achievement of the wide RC based on multiple-stacked PRs remains elusive within existing memristive frameworks as well as other device platforms due to the complex fabrication steps for 3D stacked structures and the stringent requirements on the PR properties.

For instance, many memristors show unreliable switching characteristics such as variation, instability, and low yield[21,22] owing to the intrinsic nature of switching filaments, which makes it extremely challenging to meet the high reproducibility requirement of PR states from cycle to cycle and from device to device. Additionally, the matrix array structure demands additional components, such as transistors or selectors, to ensure reliable programming and reading by preventing crosstalk[23,24]. This leads to complex device structure and fabrication process (i.e., incorporating 1 T) as well as increased power consumption and active node size. Moreover, memristors at matrix nodes must operate in a low programming voltage range because time-series data processing requires a considerable number of input pulses with varying intervals ($\Delta t$) over a specific time frame for low-power operation[14,15]. Furthermore, addressing and predicting correlations between various dynamic information sources require the ability to handle multiple time series concurrently in each time frame without mutual interference[25–27]. In other words, many independent reservoir states controlled by sequential input pulses with varying $\Delta t$ are required for multiple dynamic tasks to be performed simultaneously without interference. To effectively tackle these challenges all at once, we propose and demonstrate the wide RC based on a multilayered 3D-integrated memristive array structure with several independent reservoir states at each matrix layer, crosstalk-free and robust switching characteristics, high device yield, and low programming voltages. In other words, the primary objective of our research is to design, fabricate, and implement a wide physical RC platform, enabling its capacity to be widely applied in sophisticated reservoir configurations and providing functional benefits that go beyond simply increasing the density of devices.

Our approach can efficiently capture and process 3D local dynamic features of multiple time series by implementing a 3D stacked $3 \times 10 \times 10$ tungsten oxide ($WO_x$) array architecture. Each memristor cell is designed based on asymmetric interfacial Schottky barrier engineering to achieve crosstalk-free of signals throughout the array. The 3D stacked array exhibits highly reliable and robust operation with characteristics such as a high yield (>98%), exceptional stability (>4000 consecutive sweeps and >$10^5$ cycles), switching uniformity (~0.24%), and low operating voltage (~0.7 V). We demonstrate reliable reservoir operations under various time-dependent electrical inputs without overlapping reservoir states. As a proof of concept, we perform spatiotemporal pattern classification of biological cell positions and behavior prediction of a time-dependent chaotic Lorenz attractor. Our proposal achieves a higher and faster (~4.5 times) classification accuracy for 3D positions of a biological cell using 60% fewer memristor cells, and a tenfold reduction in prediction error for the chaotic Lorenz attractor compared with the use of a single reservoir. The proposed 3D physical RC architecture supported by reliable memristors opens an avenue for efficient and compact physical structure for reservoir computing capable of time-series processing in artificial intelligence systems.

## Results
### Three-layer stacked physical reservoirs
Figure 1a illustrates a wide RC system consisting of three reservoir layers (yellow, green, and purple) and an output layer. These reservoirs can perform a nonlinear projection of temporal inputs ($u_n(t)$) onto reservoir states ($x_n^i(t)$) per feature space via the different nodes of the reservoirs (blue circles), where $i$ and $n$ are the order of the reservoir layer ($i = 1, 2, 3$) and number of nodes ($n = 1, 2, ..., n$), respectively[9,10]. As the collective reservoir states are delivered from each reservoir layer to the output layer, the weights ($W_n^i$) between them can be iteratively updated for the desired outputs ($y_n(t)$) according to an algorithm (e.g., backpropagation, linear regression). Such learning process through the output layer can be readily executed based on a linear combination of the weighted reservoir states, thereby reducing the learning costs when compared with other time-series processing algorithms. Moreover, it has been demonstrated through an algorithmic and theoretical approach that the utilization of a wide RC system is more effective in handling multiple dynamic time information simultaneously by extracting rich local features through multiple independent reservoir layers, in comparison to a single RC system (Supplementary Fig. 1 and Supplementary Note 2)[25–27]. As shown in Figs. 1a, b, the wide RC system can be physically mapped onto a 3D stacked three-layer memristive crossbar array, where each layer acts as a reservoir (three reservoirs in total). Figure 1b shows a top-view optical microscopy image of the fabricated 3D stacked three-layer $3 \times 10 \times 10$ memristive crossbar array, where each layer consists of Pt/$WO_x$/W memristor cells at each crosspoint (300 cells). In Fig. 1b, the first, second, and third $WO_x$ layers are indicated in orange, green, and purple, respectively, confirming the formation of a 3D stacked array. Figures 1c, d show the 3D stacked $3 \times 2 \times 3$ subarray (Fig. 1c) with a magnified view of the vertically integrated cells at a crosspoint (Fig. 1d), revealing a cell line width of approximately 100 μm. To facilitate the understanding of the 3D stacked architecture, the corresponding schematics are shown in Figs. 1e, f. Array fabrication with a cell line width of 20 μm is also demonstrated using conventional photolithography (Supplementary Fig. 2). Moreover, similar switching feature at nanoscale is demonstrated, as detailed below. Figures 1g–k show cross-sectional high-resolution transmission electron microscopy (HR-TEM) images and the corresponding energy dispersive spectroscopy (EDS) results of the 3D stacked array at a crosspoint, verifying the well-defined vertical integration of the three Pt/$WO_x$/W junction structures. The detailed fabrication processes are provided in the Supplementary Information (Supplementary Fig. 3) and Methods section. Note that the benefits of our approach are further discussed in Supplementary Figs. 4, 5, Supplementary Table 1, and Supplementary Note 3, 4 of Supplementary Information.

### Electrical characteristics of the 3D physical reservoirs
Figure 2a shows a representative self-rectifying current–voltage ($I$–$V$) switching curve of a memristor in the 3D stacked $WO_x$ PRs. The $I_{ON}/I_{OFF}$

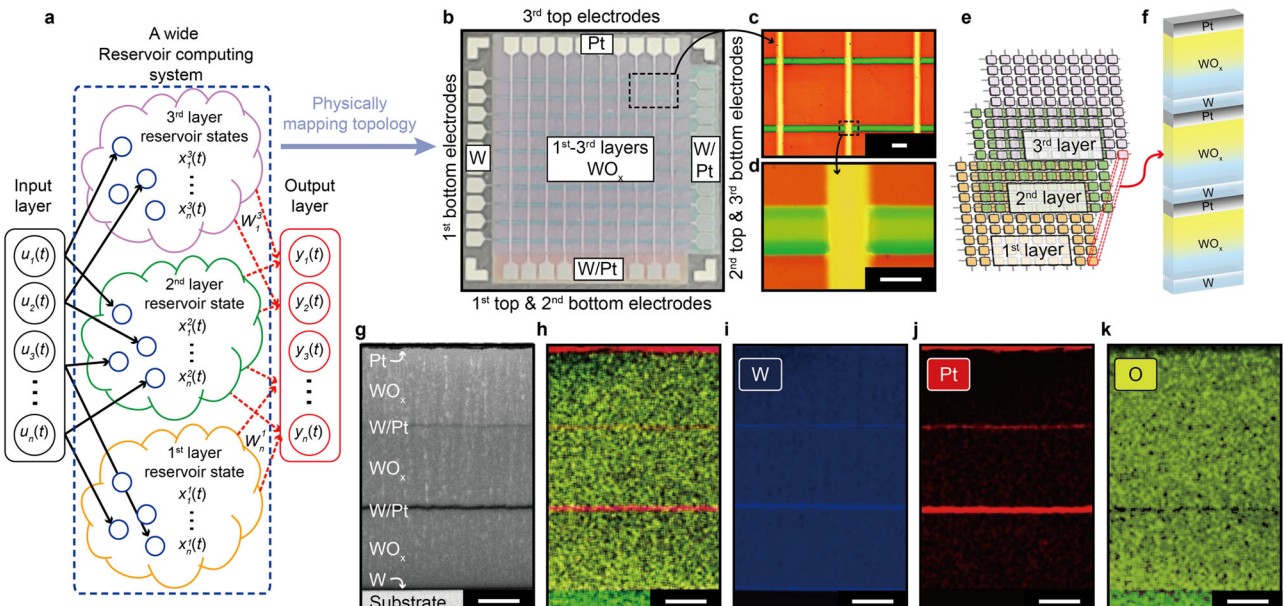

**Fig. 1 | 3D stacked WO$_x$ physical reservoir (PR) arrays. a** Schematic of a wide reservoir computing (RC) system with multiple reservoirs (see Supplementary Note 2). **b** Top-view optical image of fabricated 3D stacked three-layer $3 \times 10 \times 10$ crossbar array with vertical integration of three Pt/WO$_x$/W memristors at each crosspoint. The first (1st), second (2nd), and third (3rd) WO$_x$ layers are shown in orange, green, and purple, respectively. The fabrication of each layer is described in the Supplementary Information (Supplementary Fig. 3). The 3D stacked array can naturally map a wide RC system onto hardware. **c, d** Top-view optical image of $3 \times 2 \times 3$ sub-array (**c**) and magnified view at crosspoint (**d**) with line width of

100 μm. The scale bar for the top and bottom image is 200 μm and 100 μm, respectively. **e** Schematic of fabricated 3D stacked three-layer memristive crossbar array. Each memristor cell located at the first, second, and third layers is shown in orange, green, and purple, respectively. **f** Magnified schematic of vertical integration of three Pt/WO$_x$/W memristors at a certain crosspoint ($3 \times 1 \times 1$) in panel (**e**) (red dotted line). **g–k** Cross-sectional high-resolution transmission electron microscopy (HR-TEM) image of three stacked memristors at crosspoint (**g**) and its corresponding energy dispersive spectroscopy (EDS) results for all (**h**) and each (**i–k**) of its elements. The scale bar for all (**g–k**) is 100 nm.

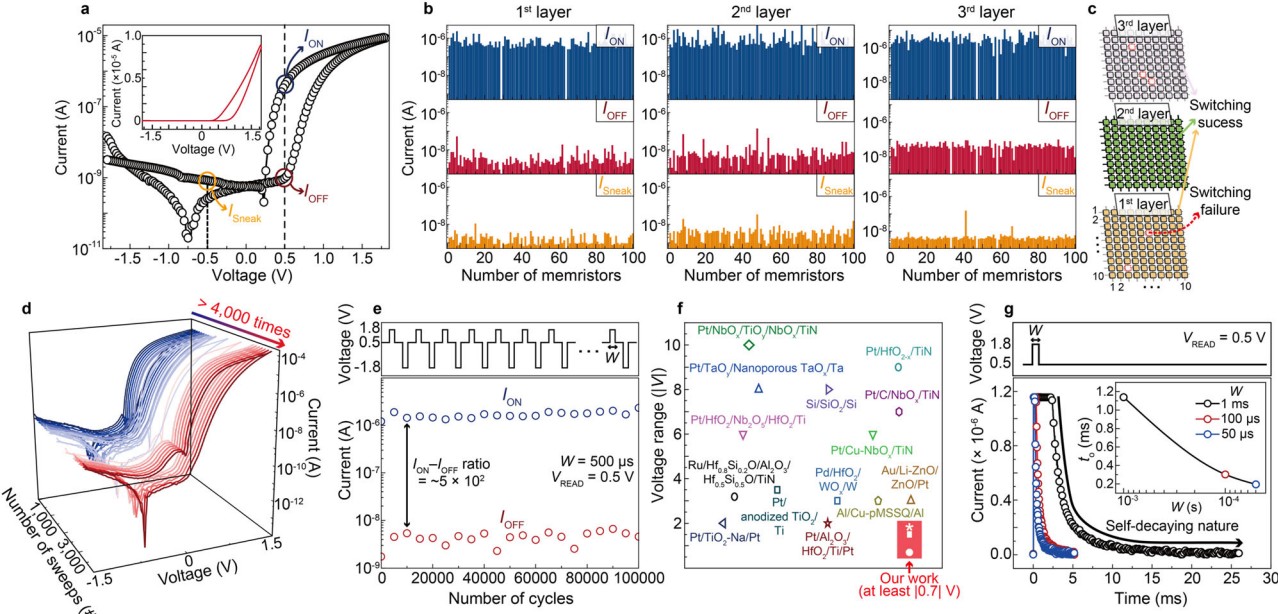

**Fig. 2 | Electrical characteristics of 3D Pt/WO$_x$/W memristor array.**
**a** Representative $I$−$V$ curve of Pt/WO$_x$/W memristor without initial forming and exhibiting self-rectifying bipolar switching. The top Pt electrode and bottom W electrode were biased and grounded, respectively. The inset shows the same $I$−$V$ curve on a linear scale. **b** Statistical histograms of switching parameters $I_{ON}$, $I_{OFF}$, and $I_{Sneak}$ for all the memristors from the first to third layers (300 memristors). **c** Schematic of 3D stacked WO$_x$ array illustrating switching success and failure (i.e., short circuit) as colored and red dotted boxes, respectively. **d** Operation under

consecutive $I$−$V$ sweeps over 4000 times for WO$_x$ memristor in 3D stacked array. **e** Endurance cycling test of WO$_x$ memristor over $10^5$ cycles by programming at 1.8 V and −1.8 V for $W$ = 500 μs. $I_{ON}$ and $I_{OFF}$ were read at $V_{READ}$ = 0.5 V. **f** Operating voltage range $|V|$ for WO$_x$ memristor (red box) and other self-rectifying memristors. **g** Dynamic $I$ behavior over time $t$ for WO$_x$ memristor after applying voltage pulses of 1.8 V for $W$ = 1 ms, 100 μs, and 50 μs, demonstrating a self-decaying nature (i.e., short-term memory) with different $t_o$ (inset).

and rectification ($I_{ON}/I_{Sneak}$) ratios are approximately $4.1 \times 10^2$ and $4.8 \times 10^2$, respectively, where $I_{ON}$ and $I_{OFF}$ are the ON and OFF currents at reading voltage $V_{READ} = 0.5$ V, respectively, and $I_{Sneak}$ is the sneak current at $-V_{READ}$. The programming voltage ranges in $\pm 1.8$ V. An initial electroforming step is not required for the fabricated array memristors. This switching phenomenon is attributed to the robust Schottky barrier formed at the Pt/WO$_x$ interface and dynamic modulation of the Schottky barrier at the WO$_x$/W interface, which is controlled by the electric-field-driven migration of oxygen vacancies[28,29]. It can be also observed even in the absence of an applied compliance current (Fig. 2a). This operational advantage can reduce the burden on peripheral circuitry for strict current control, hence, it enables our 3D stacking array to be denser and more scalable. Note that the shift of voltage position for minimum currents in Fig. 2a might be associated with the transient formation of an internal electric field[30–32] (Supplementary Note 5). Figure 2b shows the statistical histograms of $I_{ON}$, $I_{OFF}$, and $I_{Sneak}$ for all the memristors from the first to third layers of the 3D stacked WO$_x$ PRs (300 memristors). The $I$–$V$ switching curves of all the reservoir layers are provided in the Supplementary Information (Supplementary Figs. 6–8). Similar $I$–$V$ switching curves are observed regardless of the location of the reservoir layers. The average values for $I_{ON}$, $I_{OFF}$, and $I_{Sneak}$ are $(1.40 \pm 0.10) \times 10^{-6}$ A, $(4.05 \pm 0.38) \times 10^{-9}$ A, and $(2.23 \pm 0.16) \times 10^{-9}$ A, respectively. All the distributions of $I_{ON}$, $I_{OFF}$, and $I_{Sneak}$ are well-fitted by the lognormal distribution curve (Supplementary Fig. 9). Hence, well-defined switching parameters with acceptably low variations are observed in the 3D stacked WO$_x$ PRs. The device yield of the layers is 98.3% (97, 100, and 98/100 for the first, second, and third layers, respectively), as illustrated in the schematic of Fig. 2c (see Supplementary Figs. 6–8). These results support the reproducibility and expandability of the 3D stacked WO$_x$ PRs for large-scale physical RC systems.

Figure 2d, e show the consecutive switching curves for more than 4000 times and over $10^5$ cycles with an $I_{ON}/I_{OFF}$ ratio of approximately $5 \times 10^2$, respectively, demonstrating an excellent operating stability with average cycle variation of 0.24% (Supplementary Fig. 10). In addition, the multiple 3D stacked WO$_x$ PRs exhibit the lowest operating voltage range (at least |0.7 | V, Fig. 2f and Supplementary Fig. 11) with robust self-rectification (Supplementary Fig. 12), indicating the possibility of lower power consumption than existing self-rectifying memristors[31,33–45]. Figure 2g shows the current responses of the memristor cell over time after applying single voltage pulses of different widths ($W$). The current abruptly increases close to $I_{ON}$ soon after applying a single voltage pulse and then decays spontaneously and exponentially over time ($t$) under $V_{READ}$ (i.e., $I \propto exp(-t/t_o)$, where $t_o$ is the characteristic decay time), eventually returning to $I_{OFF}$. This behavior was still observed to persist despite the application of strong programming voltage scheme in both air and vacuum environments (Supplementary Fig. 13). This spontaneous decay in the switching current cannot be attributed to the protonation-assisted switching mechanism, as the metallic H$_x$WO$_3$ phase is absent and the switching occurs over a short period of time[46]. This self-decaying nature (i.e., short-term memory) can describe time-dependent internal states that enable temporal processing. Because width $W$ can control $t_o$, a 3D stacked WO$_x$ memristor cell can process time-series data over varying time frames. Note that $t_o$ was observed to decrease as the $W$ decreased based on a power function (Supplementary Fig. 14). Consequently, as compared with other reported memristor devices[47–50], our proposed WO$_x$ memristor can exhibit highly reliable selector-less short-term switching performances, operational advantages, and scalability for showcasing the 3D-integrated multilayered PR arrays.

## Switching mechanism of the 3D physical reservoirs

The interfacial barrier heights were investigated according to the $I_{ON}$ and $I_{OFF}$ states to understand the self-rectifying switching behavior of the 3D stacked WO$_x$ memristor cell. Initially, the Pt/WO$_x$/W memristor was designed based on the asymmetric distribution of oxygen vacancy ($V_O$) in WO$_x$ between Pt and W electrodes. Figure 3a shows the ex-situ X-ray photoelectron spectroscopy (XPS) results with timed Ar$^+$ bombardment for the W ion peaks at the top and bottom WO$_x$ interfaces. Compared with the top WO$_x$ interface, there are two peaks corresponding to W$^{5+}$ and W$^0$ at the bottom WO$_x$ interface (top and bottom of Fig. 3a)[51,52]. This indicates the enrichment of $V_O$ in the direction of the bottom WO$_x$ interface, which increases the concentration of $V_O$ ($C_{V_O}$) at the WO$_x$/W interface. Because the Fermi level of WO$_x$ increases (decreases) with high (low) $C_{V_O}$[53], asymmetric barrier heights $\Phi_B^1$ and $\Phi_B^2$ at the WO$_x$/W and Pt/WO$_x$ interfaces can be formed. A detailed discussion and additional XPS results are presented in Supplementary Note 5 and Supplementary Figs. 15, 16. The combined effects of the sputtering parameters and electrodes were also investigated (see the Methods section and Supplementary Figs. 17–19). The structural analysis results are consistent with the visual inspection of WO$_x$ (Supplementary Fig. 20).

Figure 3b shows the temperature ($T$)-dependent $I_{ON}$ and $I_{OFF}$ states of the Pt/WO$_x$/W memristor and their interfacial barrier heights ($\Phi_B^2$ at WO$_x$/W and $\Phi_B^2$ at Pt/WO$_x$) through the Arrhenius plot. Based on the fitting results shown in Fig. 3b (solid lines; see the Methods section), $\Phi_B^1$ and $\Phi_B^2$ are shown according to the $I_{ON}$ and $I_{OFF}$ states in Fig. 3c. The asymmetry between $\Phi_B^1$ and $\Phi_B^2$ contributes to the self-rectifying feature shown in Fig. 2a. A larger change in $\Phi_B^1$ than in $\Phi_B^2$ is observed during the switching transition, implying that charge transfer through $\Phi_B^1$ is dominant in determining the switching states. When a positive voltage is applied to the Pt electrode, $V_O$ migrates downward, and $C_{V_O}$ increases at the bottom WO$_x$ interface[28,29]. In that case, $\Phi_B^1$ is largely decreased at $V_{READ} = 0.5$ V; then, the conductance state is switched to $I_{ON}$. When a negative voltage is applied to the Pt electrode, the behaviors occur inversely, and the conductance state switches to $I_{OFF}$. Additional experimental results for different thicknesses of WO$_x$ under ambient or vacuum environments and the structural analysis of WO$_x$ are described in the Supplementary Information (Supplementary Figs. 17, 21–23, Supplementary Note 5). Note that there is a possibility that electron trapping/detrapping process may partially contribute to the switching mechanism, but it is likely to have a minor effect on the switching compared to the electric field-driven $V_O$ migration (Supplementary Fig. 24 and Supplementary Note 5).

Figures 3d and 3e show maps of the spatial current distribution ($\bar{I}$) on the Pt-tip/WO$_x$/W junction structure according to the tip bias ($V_{bias} = \pm 3$ V) using conductive atomic force microscopy (c-AFM). The estimated contact radius of the Pt tip is approximately 2.1 nm (Supplementary Fig. 25). Details of the c-AFM measurements are shown in Supplementary Fig. 25 and Methods section. The average values of $\bar{I}$ are 2.63 nA and 8.45 pA at $V_{bias}$ of 3 V and $-3$ V, respectively, demonstrating a uniform change at the whole scanning area without evident current hotspots (i.e., conductive filaments)[54]. The $I$–$V$ switching feature at the nanoscale using c-AFM is consistent with the switching behaviors observed in Fig. 2 (Supplementary Fig. 25). Figure 3f shows statistical $I_{ON}$ and $I_{OFF}$ according to the memristor size (with a constant $I_{ON}$-$I_{OFF}$ ratio, Supplementary Fig. 26), demonstrating the homogeneous enrichment and depletion of $V_O$ near the WO$_x$ interfaces[55]. Note that dependency of the $I_{ON}/I_{OFF}$ on the size is almost negligible (Supplementary Fig. 26).

## Physical reservoir computing for multiple time-series data

Temporal processing of multiple dynamic inputs was quantitatively evaluated considering the nonlinear dynamics and short-term memory of 3D stacked WO$_x$ PRs (Fig. 4 and Supplementary Figs. 27–30). Figure 4a shows the change in PR states (i.e., current responses) of the memristor cell in response at each timestep (t$_1$–t$_4$) over a voltage pulse sequence. Pulse sequence [1111] indicates that a voltage pulse ($V_{INPUT} = 1.8$ V for 200 ns) is sequentially applied at each timestep (t$_1$–t$_4$), whereas sequence [0000] indicates that no

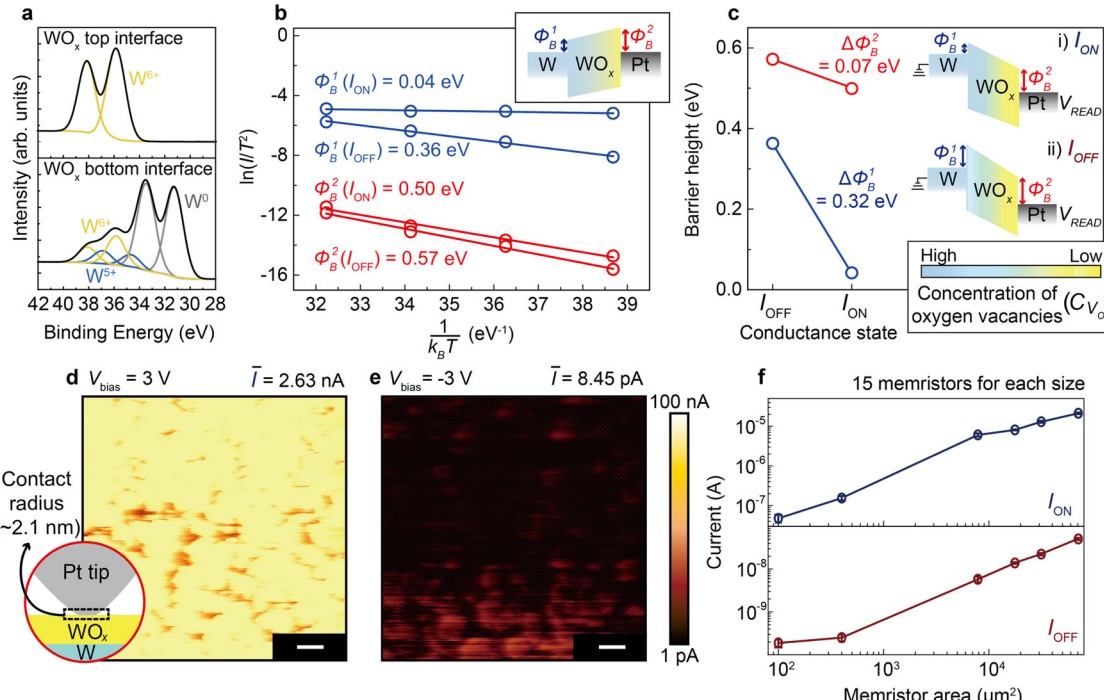

**Fig. 3 | Switching mechanism of 3D Pt/WO$_x$/W memristor. a** W 4 f XPS results taken from top and bottom interfaces of WO$_x$ layer, exhibiting asymmetric profile in concentration of oxygen vacancies ($C_{V_O}$) depending on the position of the WO$_x$ layer. **b** Arrhenius plot of ln($I/T^2$) according to $1/k_BT$ for $I_{ON}$ and $I_{OFF}$ states of Pt/WO$_x$/W memristor. Variables $\Phi_B^1$ (blue) and $\Phi_B^2$ (red) indicate each Schottky barrier height at the W/WO$_x$ (bottom) and Pt/WO$_x$ (top) interfaces, as shown in the inset. The changes in $\Phi_B^1$ and $\Phi_B^2$ were estimated from the slopes of ln($I/T^2$) according to $1/k_BT$ for the $I_{ON}$ and $I_{OFF}$ states of the Pt/WO$_x$/W memristor, respectively. **c** Changes in estimated $\Phi_B^1$ and $\Phi_B^2$ for $I_{ON}$ and $I_{OFF}$ states. The inset shows the different band diagrams for the $I_{ON}$ and $I_{OFF}$ states and state-dependent $C_{V_O}$. **d, e** Conductive atomic force microscopy (c-AFM) images of current distributions in 1 × 1 μm$^2$ WO$_x$/W for tip bias ($V_{bias}$) = 3 V (**d**) and $V_{bias}$ = −3 V (**e**). Note that spatial current distribution ($\bar{I}$) was found to be 2.63 nA (**d**) and 8.45 pA (**e**) on the average. The scale bar for both (**d**, **e**) is 100 nm. **f** Statistical $I_{ON}$ and $I_{OFF}$ values for 15 WO$_x$ memristors according to memristor area.

voltage pulses are applied. Hence, 16 reservoir states are successfully programmed at t₄ without notable cycle variations (10 cycles per sequence, Supplementary Fig. 27). These distinct reservoir states may be attributed to the self-rectifying properties and the consistent cycling switching. The trajectories of change in the reservoir states by the voltage pulse sequences are similarly observed on the memristors from the first to third layers of 3D stacked WO$_x$ PRs (Supplementary Fig. 28).

To demonstrate the capability of 3D stacked WO$_x$ PRs to process multiple time series without mutual interference, a detection task of moving biological cells was conducted based on their layer separation in nucleus, cytoplasm, and background, as shown in Figs. 4b–k[56]. Figs. 4b, c show a centered cell image (Fig. 4b) and its simplified 5 × 5-pixel image (Fig. 4c), where the blue, yellow, and black pixels represent the nucleus, cytoplasm, and background, respectively. As shown in Fig. 4d, the moving cell can be localized in one of five possible cell positions with respect to its nucleus: top left (TL), top right (TR), center (C), bottom left (BL), or bottom right (BR). In three layers of 5 × 5-pixel images corresponding to the nucleus, cytoplasm, and background images, each layer is encoded according to the five position cells using different voltage sequences in the spatial and time domains. For the centered cell (C) in Figs. 4e, f, various voltage sequences for the five WO$_x$ memristors in each layer are encoded. The multiple 3D stacked WO$_x$ PRs can individually capture three local features of the biological cell from the first to third layers. Figure 4g shows the final reservoir states at the center position in terms of these three features. The features are spatiotemporally distinguishable, demonstrating the success of feature extraction using dynamic cell positioning. The results for the other cell positions (TL, TR, BL, and BR) are shown in Supplementary Fig. 29.

When each spatiotemporal input for a cell position enters each reservoir layer, the reservoir state, $x_n^i$, in each layer ($i$ = 1, 2, 3) is generated. It is then connected to a digital output layer ($y_1(t)$, $y_2(t)$, $y_3(t)$, $y_4(t)$, and $y_5(t)$) to perform learning based on the simple single-layer gradient descent (Fig. 4h). Details about learning and prediction are provided in the Methods section. Figure 4i shows the loss function over epoch for a single reservoir and multiple reservoirs. As shown in Fig. 4i, the loss function for multiple reservoirs rapidly decreases with the number of epochs (~4.5 times faster), leading to a higher classification accuracy. Note that the further discussion on the convergence speed is described in Supplementary Note 6. In this detection task, the accuracy of the multiple 3D stacked WO$_x$ PRs is 100% at 100 epochs, being 7% higher than that of a single reservoir. Thus, for multiple PRs, only the diagonal boxes are fully saturated with yellow in the confusion matrices for the classification results between the actual and predicted positions (Fig. 4k). However, for a single reservoir, the blue saturation of several off-diagonal boxes is observed, indicating incorrect predictions (Fig. 4j). In addition, a single reservoir requires a larger number of memristors (25) than multiple reservoirs (15) for this detection task, and it cannot distinguish between the nucleus and cytoplasm at the overlapping position (Supplementary Fig. 30). Therefore, multiple PRs can extract rich features from time-series data because each reservoir learns distinct local features, even when using 60% fewer memristors than a single reservoir, thus surpassing the performance of a simple increase in packing density. Additional discussions on the wide RC are described in Supplementary Note 2, 6.

As another example of physical RC, we considered a time-dependent Lorenz attractor, which is a nonlinear and deterministic 3D chaotic dynamic system for representing convection in the atmosphere that is difficult to predict[57]. We attempted the Lorenz attractor

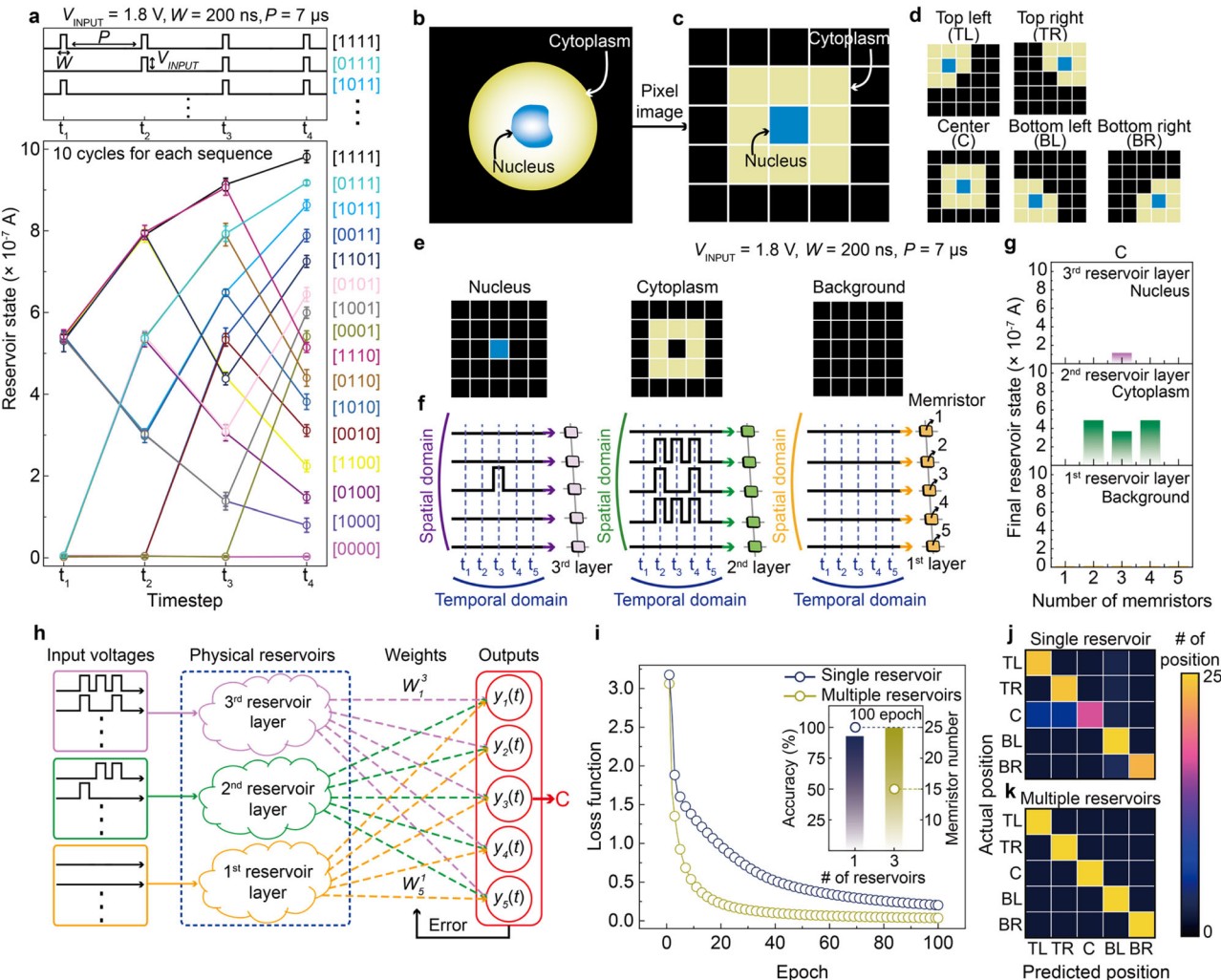

**Fig. 4 | Physical reservoir computing for cell position classification. a** Change in physical reservoir (PR) states of memristor cell at read voltage ($V_{READ}$) = 0.5 V per timestep ($t_1$–$t_4$) and 16 voltage pulse sequences ([1111], [0111], …, [0000]). A voltage pulse ($V_{INPUT}$) is 1.8 V for 200 ns with interval ($P$) = 7 μs. Each reservoir state at $t_4$ is clearly separated regardless of the cycle variation. **b, c** Illustrations of biological cell showing nucleus, cytoplasm, and background (**b**), and its corresponding 5 × 5-pixel image (**c**). **d** Schematics of five cell positions: top left (TL), top right (TR), center (C), bottom left (BL), and bottom right (BR). **e, f** Schematics of three 5 × 5-pixel images of nucleus, cytoplasm, and background for case C (**e**), and their corresponding time-series sequences applied to three PR layers (nucleus for third layer, cytoplasm for second layer, and background for first layer) (**f**). The time-series inputs were

voltage pulses converted from each local feature and introduced into different locations of the memristors in each layer (5 memristors per layer for the spatial domain) at different times ($t_1$–$t_5$ in the temporal domain). **g** Final reservoir states at $t_5$ obtained from first to third reservoir layers (5 memristors per layer). **h** Schematic of reservoir computing (RC) configuration consisting of input layer for time-series data, three PRs, and output layer. Weights ($W_n^i$) between PRs and output layer was trained to reduce the error (i.e., loss function) by comparing outputs ($y_n(t)$) and the desired value. **i** Loss functions for single (navy) and multiple (yellow) reservoirs according to training epoch. The inset shows the classification accuracies at 100 epochs and numbers of used memristors for the single and multiple reservoirs. **j, k** Confusion matrices for single (**j**) and multiple (**k**) reservoirs.

prediction and simulation using multiple 3D stacked WO$_x$ PRs as a proof of concept. Figure 5a shows the 3D trajectory obtained from the Lorenz attractor equations with specific parameters (see the Methods section). As shown in Figs. 5b, c, by deconvoluting the 3D Lorenz equation into three time-dependent one-dimensional equations (i.e., x-, y-, and z-axis components over a timestep) (Fig. 5b), the 3D stacked WO$_x$ PRs could directly handle their corresponding chaotic inputs to generate separable reservoir states, which were then transferred to the output layer (Fig. 5c).

Based on the linear regression algorithm, the learning process was performed at the output layers considering 100 nodes per reservoir (300 nodes in total). As shown in Fig. 5d, after initialization and learning over 1400 timesteps, the predicted behavior closely matches the actual behavior with a negligible deviation. Figure 5e shows the amplitudes of the x, y, and z components over time for the actual and predicted behaviors (2788 timesteps). Hence, we demonstrate the

successful prediction of a time-dependent 3D Lorenz attractor using multiple 3D stacked $3 \times 10 \times 10$ WO$_x$ PRs. The average normalized mean squared error (NMSE) between the predicted and actual behaviors is approximately $2.62 \times 10^{-4}$, being >10 times better than that when using a single reservoir with 300 nodes (NMSE of $1.35 \times 10^{-3}$) (Fig. 5f and Supplementary Fig. 31). If the reservoir capacity is insufficient, even when considering multiple reservoirs, the predicted behavior substantially deviates from the actual behavior (Supplementary Fig. 32). Details about prediction are provided in the Methods section. The results indicate that the optimized $W_n^i$ in the output layer obtained from learning allows to suitably predict the future behavior in response to multiple reservoir states (three layers) at the current timestep. As a result, 3D stacked WO$_x$ memristor arrays promote the prediction accuracy and efficiency of physical RC for multiple time-series processing. Note that, in Supplementary Note 6 and Supplementary Figs. 33, 34, we show relevant discussions of physical RC

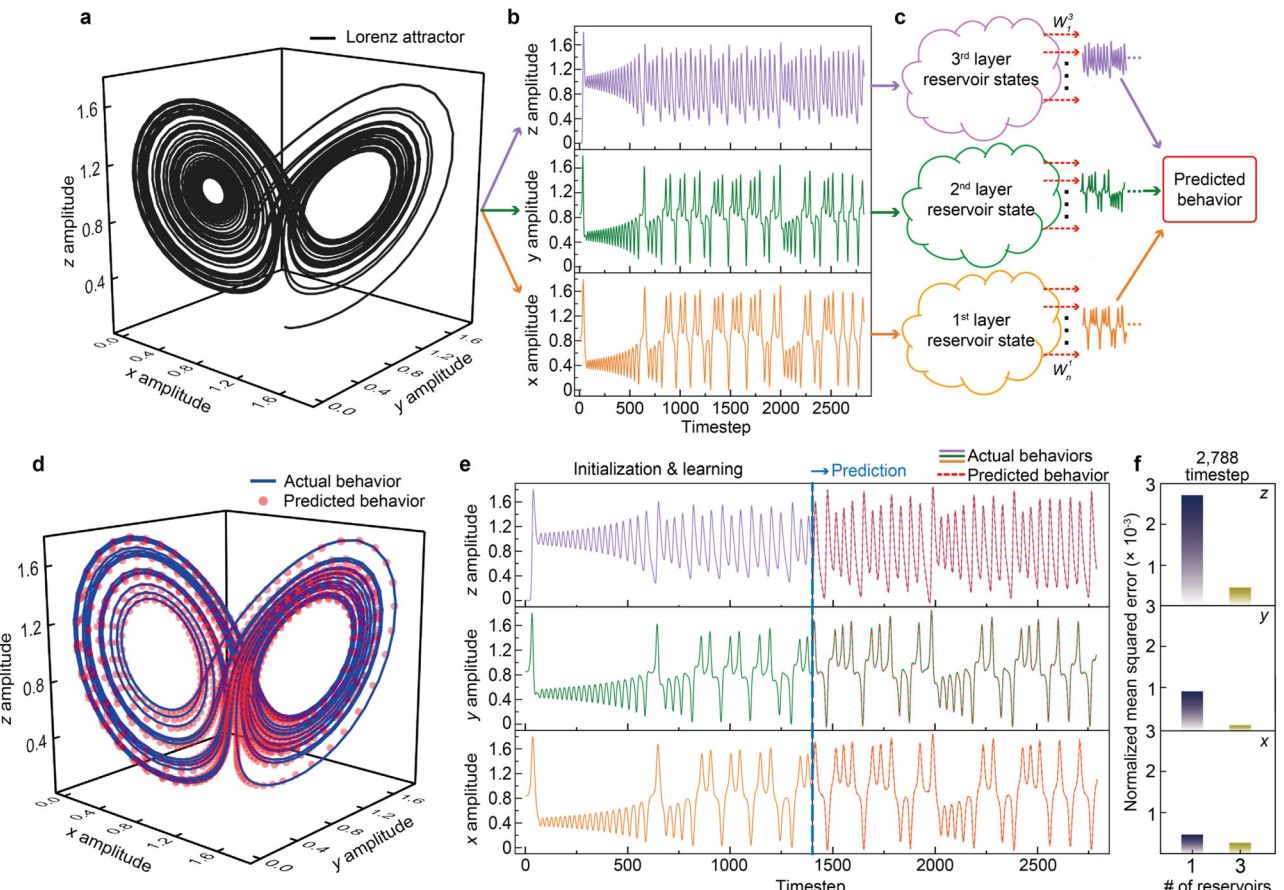

**Fig. 5 | Physical RC for prediction of time-dependent Lorenz attractor. a** 3D trajectory of Lorenz attractor in the *x*, *y*, and *z* components. **b**, **c** One-dimensional time-dependent Lorenz attractors over time deconvoluted from 3D Lorenz attractor (**b**), and schematic of three physical reservoirs (PRs) for prediction of the 3D Lorenz attractor (**c**). **d** Actual and predicted behaviors of 3D Lorenz attractor after 1400 timesteps. **e** Individual actual behaviors of *x*, *y*, and *z* components over time (colored solid lines), and their corresponding prediction results (red dotted lines) using multiple 3D stacked 3 ×10 × 10 WO$_x$ PRs after 1400 timesteps (blue dotted line). **f** Normalized mean squared error (NMSE) between actual and predicted behaviors at timestep 2788 for single (navy) and multiple (yellow) reservoirs in the *x*, *y*, and *z* components.

based on multiple 3D stacked WO$_x$ PRs as compared with traditional approaches and suggest how they could be used at the circuit level.

## Discussion

We design and fabricate a 3D stacked three-layer $3 \times 10 \times 10$ WO$_x$ memristive crossbar array to implement a wide physical reservoir system for processing multiple time-series information. This design exhibits a forming-free feature, high switching uniformity and device yield (>98%) for a three-layer stacked array, robust endurance (>10$^5$ cycles), and low programming voltage (-0.7 V) for self-rectifying switching. Moreover, it demonstrates essential reservoir properties, including separation and memory fading. The 3D stacked WO$_x$ memristive crossbar array for wide physical RC allows each array layer to individually portray and process distinct local features of spatio-temporal inputs such that the reservoir states can be readily analyzed at the output layer for classification and prediction tasks. By leveraging this strategy, for dimensional classification of biological cell positions, the multiple PRs achieved better classification accuracy (~100%) and ~4.5 times faster convergence using 60% fewer memristors (15) compared with a single reservoir (93% accuracy with 25 memristors). In addition, for the prediction of the chaotic Lorenz attractor, they achieved a tenfold lower prediction NMSE (-2.62 × 10$^{-4}$) than a single reservoir (NMSE of 1.35 × 10$^{-3}$). Although the array size is still small as a proof-of-principle demo, both the principles and device structure are applicable to large scales. The proposed 3D stacked WO$_x$ PR array

represents a leap forward toward an efficient wide RC platform, promising for time-series processing in edge computing.

## Methods

### Fabrication of 3D stacked three-layer memristive crossbar array

The 3D stacked three-layer $3 \times 10 \times 10$ WO$_x$ memristive crossbar array was fabricated on either a glass substrate or 285 nm-thick SiO$_2$/p$^{++}$-Si substrate (1.5 cm × 1.5 cm). First, the substrates were prepared using a conventional cleaning process with acetone, isopropyl alcohol, and deionized water via ultra-sonication for 3 min each. To form 1st bottom electrode lines (< -10 nm thick) on the substrate, W was deposited by sputtering system with 5 min pre-sputtering step, gas flow of Ar 10 sccm, working pressure of 10 mTorr, and gun power of 100 W with the patterned shadow mask (1.5 cm × 1.5 cm) with line width of 100 μm. Note that all the following fabrication processes were performed through sputtering and shadow masks to circumvent the sidewall issue that is critical for the 3D stacked structure. The same-sized shadow masks with the substrate were used to facilitate the patterning alignments. Subsequently, the WO$_x$ (-180 nm thick) was deposited on the center of the 1st bottom W electrodes except for the contact pads. The patterning alignment was carefully carried out with an optical microscopy. During WO$_x$ deposition, various combinational sputtering parameters were used, such as 20 min pre-sputtering step, gas flow of Ar 60 sccm and O$_2$ 1 sccm, working pressure of 20 mTorr, gun power of 150 W, and substrate temperature of 40 °C, to optimize a nonlinear

dynamic switching with self-rectification (see Supplementary Fig. 17). Perpendicularly to the 1st bottom W electrode lines, the 1st top Pt electrode lines (< -10 nm) were formed on the 1st $WO_x$ layer by using a shadow mask and sputter. The Pt was deposited with 5 min pre-sputtering step, gas flow of Ar 10 sccm, working pressure of 10 mTorr, and gun power of 100 W. After the Pt deposition, W was directly sputtered on the 1st top Pt electrodes to form 2nd bottom W electrode lines. The 2nd $WO_x$ layer was then formed on the 2nd bottom W electrode/1st top Pt electrode lines, except for the all electrode pads. The same sputtering parameters used during the formation of 1st $WO_x$ were employed. Similar fabrication steps were repeated until completing the 3D stacked of the three-layer memristive crossbar array. The top-view optical images of the fabrication processes and corresponding schematics are shown in Supplementary Fig. 3.

### Structural analysis
To identify the formation of 3D stacked junction structure at a certain crosspoint, the sample for TEM and EDS analysis was prepared via focused ion beam technique (FIB, Helios G4 HX) and then cross-sectional TEM image and EDS results of the 3D stacked memristors were obtained by using a FEI Double Cs and monochromated TEM instrument. The depth-profiling XPS analysis (PHI 5000 VersaProbe) was performed by using 100 μm-diameter sized single Pt/$WO_x$/W memristor (80-nm thick $WO_x$).

### Electrical characterization
A semiconductor parameter analyzer (4155 C, Agilent), pulse generator (81104 A, Keysight), and low-leakage switch mainframe (E5250A, Keysight) were used to electrically investigate and characterize the $I-V$ switching curves, consecutive sweeps, endurance cycles, voltage ranges, temperature-dependent Schottky barrier height, and size dependency of the 3D stacked three-layer $3 \times 10 \times 10$ $WO_x$ memristive crossbar array. Memristors in the 3D stacked arrays were randomly accessible when contacting their corresponding electrode pads. For example, to select a certain memristor of the 1st layer, the 1st bottom W electrode pad was grounded, whereas the 1st top Pt & 2nd bottom W electrode pad was biased. Considering that the 1st top Pt electrode is underneath the 2nd bottom W electrode, the Pt/$WO_x$/W memristor of the 1st layer can be characterized. This measurement can be considered a reasonable method because our $WO_x$-based memristors operate based on interfacial switching. The time-decay features and current responses (i.e., reservoir states) as functions of various time-dependent inputs were measured by using a Keithley 4200 semiconductor characterization system that can program arbitrary voltage schemes into memristors. To investigate self-decaying nature, a $V_{READ}$ of 0.5 V for 5 μs was first applied into the selected memristor in the 3D stacked array to confirm $I_{OFF}$ state. Then, a programming pulse with amplitude of 1.8 V for pulse widths $(W) = 1$ ms, 100 μs or 50 μs, which can switch the device into $I_{ON}$ state, was introduced into the memristor. The rising and falling edges of the pulses were set to be 500 ns. After the programming, a $V_{READ}$ of 0.5 V was used to investigate the current decaying behaviors over time. To generate reservoir states, time-varying voltage pulse schemes consisting of four timesteps were used (Fig. 4a). As shown in the upper panel of Fig. 4a, 16 binary encoding sequences (1 or 0) consist of 4 timesteps ($t_1$–$t_4$) at $P = 7$ μs, and each timestep contains either a $V_{INPUT}$ of 1.8 V for $W = 200$ ns when its corresponding encoding is 1 or no pulse when its corresponding encoding is 0. The rising and falling edges of the programming pulse were set to be 50 ns. After each timestep, $V_{READ}$ of 0.5 V at the time interval of 5 ns was used to record reservoir states (i.e., current behaviors). Note that for the random access, the same method used in the electrical characterization was employed.

### Conductive atomic force microscopy
To identify the filament-free and homogeneous switching phenomena of the $WO_x$ memristor, conductive atomic force microscopy (c-AFM)

measurement was performed. In order to map the spatial current distribution ($\bar{I}$), a Pt/Ir-coated c-AFM tip was used to directly scan the top surface of the $WO_x$/W sample and configure the Pt/$WO_x$/W junction structure (Supplementary Fig. 25). The Pt/Ir-coated tip was set to be grounded, and the bottom W electrode was set to be biased; hence, tip bias ($V_{bias}$) was represented as the opposite bias polarity. The tip (~24 nm) and estimated contact radii (~2.1 nm) are discussed in the Supplementary Information (Supplementary Fig. 25). The electrical characteristics were measured in scanning mode with a scan area of $1 \times 1$ μm² by using a DLPCA-2000 built-in current amplifier (Electro-Optical Components) under a humidity below 15%. After the Pt/Ir tip contacted the $WO_x$/W samples, −3 V and 3 V were applied to the bottom W electrode during scanning, respectively. Then, the spatial current distribution ($\bar{I}$) at $V_{bias} = 3$ V and −3 V were mapped on the $1 \times 1$ μm² scan area and it was verified that there is no occurrence of a conductive filament.

### Estimation of Schottky barrier height
To determine the Schottky barrier heights of the Pt/$WO_x$/W memristor, temperature-dependent $I-V$ curves were investigated and plotted according to the following thermionic emission equations:

$$I = AA^*T^2 \exp\left(\frac{-q\Phi_B}{K_B T}\right), \tag{1}$$

$$ln\frac{I}{T^2} = \ln(AA^*) - \frac{q\Phi_B}{K_B T}, \tag{2}$$

where $A, A^*, T, q, \Phi_B$, and $K_B$ are the effective area, Richardson constant, temperature, electronic charge, Schottky barrier height, and Boltzmann constant, respectively. Hence, $\Phi_B^1$ and $\Phi_B^2$ for the ON and OFF states at $V_{READ} = \pm 0.5$ V are quantitatively estimated from the slope of the straight fitting lines shown in Fig. 3b.

### Dimensional classification of biological cell positions
The dimensional classification of five biological cell positions was simulated based on the individual PR states encoded from each layer of the 3D stacked $WO_x$ array. Three layers of $5 \times 5$-pixel images corresponding to the nucleus, cytoplasm, and background images are individually encoded by each layer of the 3D stacked $WO_x$ array. A Keithley 4200 system, capable of generating arbitrary voltage pulse patterns, was employed to measure the reservoir states (i.e., current responses) at each timestep (from $t_1$ to $t_5$) by applying the spatio-temporal inputs shown in Figs. 4e–g. Five memristors per layer were selected in each $10 \times 10$ crossbar array for spatial information processing of the biological cell, and all reservoir states were read at $V_{READ}$ past 5 μs after the application of each programming pulse (1.8 V for 200 ns). The programming and reading disturbances in the 3D stacked $3 \times 10 \times 10$ $WO_x$ array can be minimized owing to the self-rectifying function. The reservoir states of the different memristors in each layer were individually recorded by applying time-dependent voltage pulses to the top Pt electrode line of the selected memristor with a grounded bottom W electrode line.

Subsequently, a single-layer neural network with a supervised learning algorithm was employed as the output layer. By considering the PR states at the final time frame ($t_5$) as input $x$, output $y$ was generated through the softmax activation function:

$$y_j = \frac{e^{z_j}}{\sum_{k=1}^j e^{z_k}}, z_j = W_{i,j} \cdot x_i + b, \tag{3}$$

where $W_{i,j}$ is the synaptic weight between a reservoir and output layer, $b$ is the bias, and $i$ and $j$ are integers identifying the reservoir nodes and output neurons, respectively. The number of output neurons was five, corresponding to the considered cell positions (top left (TL), top right

(TR), center (C), bottom left (BL), or bottom right (BR)). During learning, the cross-entropy loss function ($\varepsilon$) was decreased using delta rule of gradient descent as follows:

$$\varepsilon = -\frac{1}{n}\sum_{m=1}^{n}\sum_{k=1}^{j}\hat{y}_k^{(m)}\log(y_k^{(m)}), \qquad (4)$$

$$W_{i,j} \equiv W_{i,j} - \eta\frac{\partial\varepsilon}{\partial W}, \qquad (5)$$

where $n$ is the number of input data points, $\hat{y}_j$ is the one-hot encoding of $y_j$, and $\eta$ is the learning rate. After learning, additional images were generated per cell position and used during prediction (Supplementary Fig. 35), and also their corresponding PR states were produced based on the fitting results (Supplementary Fig. 36).

### Prediction of time-dependent Lorenz attractor

The 3D Lorenz attractor is a set of chaotic solutions of the Lorenz system described by three ordinary differential equations according to the axis:

$$\frac{dx}{dt} = \alpha(y - x), \qquad (6)$$

$$\frac{dy}{dt} = x(\beta - z) - y, \qquad (7)$$

$$\frac{dz}{dt} = xy - \gamma z. \qquad (8)$$

These equations can describe various chaotic behaviors and physical phenomena over time depending on the values of parameters $\alpha$, $\beta$, and $\gamma$. By setting $\alpha = 10$, $\beta = 28$, and $\gamma = 8/3$ for the chaotic behaviors, the Lorenz attractor was normalized and deconvoluted into time-varying amplitudes for the $x$, $y$, and $z$ components. These values were applied into each PR after normalizing the inputs in the operating voltage range from 0 V to 1.8 V, as shown in Figs. 5b, c. At each reservoir, 100 memristors were utilized and simulated to enrich the reservoir states because some device variations can produce qualitatively similar but quantitatively different current responses (Supplementary Fig. 28 and Supplementary Note 2, 6). The reservoir states obtained from the three PRs were fed into the corresponding output layers, and the weights were optimized using the supervised learning algorithm of linear regression with the pseudo inverse:

$$W = \left(X^{\mathsf{T}}X\right)^{-1}X^{\mathsf{T}}Y \qquad (9)$$

$X$, $W$, and $Y$ are a set of reservoir states, weights, and actual behaviors, respectively. After initialization during the initial 100 timesteps, the weights were updated until 1400 timesteps. Initialization was required to prepare and assist the prediction processing of the RC system without learning because the initial conditions of the Lorenz attractor could strongly affect its future behavior. When learning was complete, the actual behaviors at the current timestep ($t$) were used to predict the behaviors at the next timestep ($t+1$).

### Normalized mean squared error

The normalized mean squared error (NMSE) used for performance evaluation was calculated as

$$\mathrm{NMSE} = \frac{\sum_t\sum_j(y_j(t) - t_j(t))^2}{\sum_t\sum_j t_j^2(t)}, \qquad (10)$$

where $t_j(t)$ and $y_j(t)$ are the actual and predicted behaviors, respectively.

## Data availability

All data supplementary to the findings of this study are available within the article and its Supplementary Information, or from the corresponding author upon request.

## Code availability

All codes used for the simulation are available within the article and its Supplementary Information, or from the corresponding author upon request.

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

## Acknowledgements

This work was supported by the National Research Foundation of Korea (NRF-2022R1A2B5B02001455, NRF-2022M3H4A1A01009526, NRF-2022K2A9A1A01098066, and RS-2023-00220077), the Korea Institute of Science and Technology (KIST) Institutional Program (Project No. 2E32491-23-112), the KU-KIST Research Fund, and a Korea University grant.

## Author contributions

G.W. and S.C. conceived the research and wrote the manuscript using inputs from all the authors. S.C. performed the experiments, analyses, and neural network simulations. J.S. and J.S.E. performed the c-AFM measurements on the $WO_x/W$ sample. G.P. and J.J. assisted in the execution of some experiments, such as $WO_x$ deposition and electrical characterization. J.J.Y. participated in the discussions regarding the experimental results and provided an interpretation of them. G.W. oversaw the project, revised the manuscript, and led the efforts toward completion with J.J.Y.

## Competing interests

The authors declare no competing interests.

## Additional information

**Supplementary information** The online version contains Supplementary Material available at https://doi.org/10.1038/s41467-024-46323-7.

