## [Peer Review File · Nature Communications]

Reviewers' Comments:

Reviewer #1:

Remarks to the Author:

The authors have carefully addressed all my concerns and amended and improved the manuscript. I do not have additional comments and recommend the manuscript for publication in its present form.

Reviewer #2:

Remarks to the Author:

I appreciate the authors' timely responses, which have effectively addressed my previous concerns. I just have a minor suggestion this time: please provide a comparison between the training cost reduction and energy efficiency of the two demonstrations and conventional digital hardware running standard deep learning models. This helps to illustrate the advantages of physical reservoirs, which offer cost-effective training and high efficiency.

Reviewer #3:

Remarks to the Author:

The author has addressed all the reviewers' concerns satisfactorily. I agree its publication.

Response to Reviewer #1

[Comments]

The authors have carefully addressed all my concerns and amended and improved the manuscript. I do not have additional comments and recommend the manuscript for publication in its present form.

RESPONSE: We appreciate the reviewer's comment. Again, we believe that the quality of our manuscript was largely enhanced by the insightful remarks and comments made by reviewer #1.

Response to Reviewer #2

[Comments]

I appreciate the authors' timely responses, which have effectively addressed my previous concerns.

RESPONSE: We sincerely appreciate you for taking the time to evaluate our revised manuscript. Furthermore, we extend our gratitude to reviewer #2 for providing insightful feedback that significantly enhanced the quality of our manuscript and for approving its publication.

#1. I just have a minor suggestion this time: please provide a comparison between the training cost reduction and energy efficiency of the two demonstrations and conventional digital hardware running standard deep learning models. This helps to illustrate the advantages of physical reservoirs, which offer cost-effective training and high efficiency.

RESPONSE: We are grateful for the reviewer's feedback. As the reviewer said, we concur that this comparison can serve to underscore the benefits associated with physical reservoirs (PRs).

To address the reviewer's comment, we have performed the prediction of Lorenz attractor via one of the conventional standard deep learning models, 100-1000-1000-1 multilayered perceptron (MLP) consisting of an input, two hidden, and an output layer (Fig. R1). Here, as a good comparator, we have only considered the prediction of Lorenz attractor data. This is because this task has significantly more timesteps than biological cell images, which can show the stark difference in efficiency between the two platforms. In this simulation, similar to the configuration of wide physical reservoir computing (RC), we employed three MLPs for predicting x , y , and z components of Lorenz attractor (Fig. R1a). Moreover, because the input at a certain timestep is not stored and is independent of each other in typical MLPs, inputs at the current timestep (t) and the previous timesteps ($t-1$, $t-2$..., $t-99$) were simultaneously fed to the networks during learning processes (Fig. R1b), where loss function, learning rule, learning epoch, and batch size are mean square error, RMSprop (learning rate = 0.01 and exponential decay=0.9), 200, and 2, respectively. The activation function of the hidden and output layers was set to rectified linear unit (ReLU) and linear function with bias, respectively. After learning processes until 1400 timesteps, as shown in Figs. R1c and R1d, predicted behavior largely deviates the actual behavior. The average normalized mean square errors (NMSEs) were estimated to be $\sim 9.46 \times 10^{-2}$. This result is >300 times worse than that of the wide physical RC ($\sim 2.62 \times 10^{-4}$), although the MLP approach has the much larger number of learning parameters (i.e., $1,103,001 \times 3 = 3,309,003$) than that of the wide physical RC ($100 \times 3 = 300$). This might be attributed to the insufficient capability to capture certain nonlinear dependencies between sequential incoming inputs. Similarly, for the prediction of chaotic Mackey-glass equation, single physical RC has already shown better prediction results than deep learning models such as MPL and long short-time memory (LSTM) with much fewer learning parameters (Moon et al., Nat. Electron. 2, 480 (2019)). Consequently, these results support that physical RC can efficiently process time-series data at lower learning cost.

Fig. R1. a, A schematic of three MLPs for prediction of the 3D Lorenz attractor. **b,** An example of 100-1000-1000-1 network configuration for the 3rd MLP. Note that bias is omitted for simple illustration. **c,** Actual and predicted behaviors of 3D Lorenz attractor. **d,** Individual actual behaviors of x , y , and z components over time (colored solid lines), and their corresponding prediction results (red dotted lines) using the three MLPs after 1,400 timesteps (blue dotted line). The prediction of the Lorenz attractor was unsuccessful, which might be related to the insufficient capability of capturing temporal correlation between sequential inputs (Supplementary Note 6).

In addition to this, we have also compared the wide physical RC with a digital computer running software-based wide RC in terms of the energy efficiency for the same task. This comparison based on similar pre-conditions would be appropriate to appeal to the advantage of the physical RC system.

In fact, in the first-round revision, we have already examined the energy consumed during the prediction of Lorenz attractor for i) 3D stacked PR arrays and ii) software-based multiple reservoirs built on a digital computer based on a few assumptions for simple calculation (the comment #8 of Reviewer #3). Briefly, we estimated the energy (E_{node}) consumed at each software-based reservoir node and each input using a software-based real-time power monitoring program based on the Running Average Power Limit (RAPL) interface, which was found to 3.05×10^{-5} J. For the 3D stacked physical reservoir array, the energy (E_{mem}) at each physical reservoir node and each input could be estimated by $E_{mem} = V_{INPUT} \times I \times W$, where V_{INPUT} , I , and W are input voltage, current response, and pulse width, respectively. When the V_{INPUT} and I were set to be 1.8 V and $\sim 10^{-5}$ A to represent their maximum possible values, the E_{mem} was increased from $\sim 3.60 \times 10^{-12}$ J to 1.80×10^{-8} J as the W increased from 200 ns to 1 ms. Based on this comparison, we can say that the reservoirs of the 3D-integrated multilayer PR array could decrease the consumed energy by up to $\sim 8.47 \times 10^6$ times compared to that of the conventional computing system. Therefore, we believe that this result supports the aim of implementing a physical wide RC system based on the 3D stacked PR array to efficiently process multiple dynamic time-series information.

In response to the reviewer comment, we revised and added the following paragraphs in the revised manuscript.

- Note that, in Supplementary Note 6 and Figs. S33 and S34, we show relevant discussions of physical RC based on multiple 3D stacked WO_x PRs as compared with traditional approaches and suggest how they could be used at the circuit level. (page 11)

We added the following paragraphs in Supplementary Note 6 of the revised Supplementary Information

- As shown in Fig. S33, we have performed the prediction of Lorenz attractor via one of the conventional standard deep learning models, 100-1000-1000-1 multilayered perceptron (MLP) consisting of an input, two hidden, and an output layer (Fig. S33). Here, as a good comparator, we have only considered the prediction of Lorenz attractor data. This is because this task has significantly more timesteps than biological cell images, which can show the stark difference in efficiency between the two platforms. In this simulation, similar to the configuration of wide physical reservoir computing (RC), we employed three MLPs for predicting x , y , and z components of Lorenz attractor (Fig. S33a). Moreover, because the input at a certain timestep is not stored and is independent of each other in typical MLPs, inputs at the current timestep (t) and the previous timesteps ($t-1, t-2 \dots, t-99$) were simultaneously fed to the networks during learning processes (Fig. S33b), where loss function, learning rule, learning epoch, and batch size are mean square error, RMSprop (learning rate = 0.01 and exponential decay=0.9), 200, and 2, respectively. The activation function of the hidden and output layers was set to rectified linear unit (ReLU) and linear function with bias, respectively. After learning processes until 1400 timesteps, as shown in Figs. S33c and S33d, predicted behavior largely deviates the actual behavior. The average normalized mean square errors (NMSEs) were estimated to be $\sim 9.46 \times 10^{-2}$. This result is >300 times worse than that of the wide physical RC ($\sim 2.62 \times 10^{-4}$), although the MLP approach has the much larger number of learning parameters (i.e., $1,103,001 \times 3 = 3,309,003$) than that of the wide physical RC ($100 \times 3 = 300$). This might be attributed to the insufficient capability to capture certain nonlinear dependencies between sequential incoming inputs. Similarly, for the prediction of chaotic Mackey-glass equation, single physical RC has already shown better prediction results than deep learning models such as MPL and long short-time memory (LSTM) with much fewer learning parameters.^{S17} Consequently, these results support that physical RC can efficiently process time-series data at lower learning cost.

We also added Fig. R1 as Fig. S33 in the revised Supplementary Information.

Thank you once again for your valuable time and effort that the reviewer has spent on evaluating our revised manuscript.

Response to Reviewer #3

[Comments]

The author has addressed all the reviewers' concerns satisfactorily. I agree its publication.

RESPONSE: We appreciate the valuable time and effort that the reviewer has spent on evaluating our revised manuscript. Furthermore, we extend our gratitude to reviewer #3 for providing insightful feedback that significantly enhanced the quality of our manuscript and for approving its publication. Thank you once again.

Reviewers' Comments:

Reviewer #2:

Remarks to the Author:

The authors addressed all the reviewer's concerns in detail and I would like to recommend this work for publication.